# A Pilot Study on the Contamination of Assistance Dogs’ Paws and Their Users’ Shoe Soles in Relation to Admittance to Hospitals and (In)Visible Disability

**DOI:** 10.3390/ijerph18020513

**Published:** 2021-01-10

**Authors:** S. Jasmijn Vos, Joris J. Wijnker, Paul A. M. Overgaauw

**Affiliations:** Department of Population Health Sciences, Institute for Risk Assessment Sciences, Faculty of Veterinary Medicine, Utrecht University, P.O. Box 80178, 3508 TD Utrecht, The Netherlands; s.j.vos@students.uu.nl (S.J.V.);

**Keywords:** assistance dog, *Clostridium difficile*, Enterobacteriaceae, hospital, hygiene, paw, shoe sole

## Abstract

(1) Background: People with disabilities may benefit from an assistance dog (AD). Despite regulations that prohibit the denial of ADs to public places, this still occurs on a regular basis. The main argument for denial of access is that dogs compromise hygiene with their presence, which could cause a health hazard. Meanwhile, people are allowed to walk into and out of public places freely. (2) Objective: As a pilot study, to investigate the number of Enterobacteriaceae and the presence of *Clostridium difficile* bacteria on the paws of ADs and pet dogs (PDs) as well as the shoe soles of their users and owners. With the results, an assessment can be made as to whether measures are required to reduce environmental contamination (e.g., in hospitals). (3) Methods: In total, 25 ADs, 25 PDs, and their 50 users/owners participated in the study. Each participant walked their dog for 15–30 min prior to the sampling of the front paws. Each PD owner or AD user filled out a general questionnaire about the care of their dogs, and AD users were asked to fill out an additional questionnaire on their experiences regarding the admittance of their ADs to public places (in particular, hospitals). Dutch hospitals were questioned on their protocols regarding the admittance of ADs and their visitor numbers, including the percentage of AD users, to put these numbers into perspective. (4) Results: Dog paws were more often negative for Enterobacteriaceae compared to shoe soles (72% and 42%, respectively) and also had significantly lower bacterial counts (mean of 3.54log_10_ and 5.03log_10_ colony-forming units (CFUs), respectively; *p* < 0.05). This was most distinct in the comparison between PDs and their owners (3.75log_10_ and 5.25log_10_ CFUs; *p* < 0.05); the numbers were similar between ADs and their users (3.09log_10_ and 4.58log_10_ CFUs; *p* = 0.2). *C. difficile* was found on one (4%) AD user’s shoe soles. Moreover, 81% of AD users had been denied access with their current AD once or several times, the main reason being hygiene. The results of the visibly and invisibly disabled were significantly different. The number of AD users as opposed to the total number of hospital visitors was 0.03% in one hospital and is estimated to be 0.02% in the Netherlands. (5) Conclusions: The general hygiene of dogs’ paws is far better than that of shoe soles, mostly demonstrated by the better general hygiene of PD paws compared with their owners’ shoe soles; ADs and their users had comparable levels of general hygiene. In addition, the number of AD users amongst the total number of hospital visitors in the Netherlands is very limited. Thus, hygiene measures to reduce any contamination due to dog paws do not seem necessary.

## 1. Introduction

In the Netherlands, many people with a mental or physical disability benefit from the daily use of an assistance dog (AD). Thirteen accredited organisations deliver hundreds of ADs per year [1,2,3], and the largest organisation had 724 active human–dog teams in 2019 [4]. These dogs are able to guide people with a visual impairment through the world, to complete tasks for those who cannot and offer support in cases of post-traumatic stress disorder (PTSD) or autism, for example. However, the degree of acceptance differs per country and culture, which presents itself in the denial of access for these dogs to healthcare facilities, public transport, shops, and so on. It is thought to be especially the case for individuals with invisible disabilities, such as PTSD. Research conducted on behalf of The Royal Dutch Guide Dog Foundation in 2019 showed that 40% of their AD users [5] had been denied access to a public place in the past year [6]. The argument that is most frequently posited is that dogs contaminate their environment, which could be a health hazard. Meanwhile, people are allowed to walk into and out of—for example—hospitals freely, wearing the same shoes outside as well as inside.

When it comes to ADs, several different terms are used. By Dutch law and the global authority Assistance Dogs International (ADI), “assistance dog” is applied as the umbrella term. AD training foundations do not always follow this decision and sometimes create their own names for their ADs. Subgroups of ADs are guide dogs (for the guiding of people with a visual impairment), hearing dogs (for the alerting of people with a hearing impairment to specific sounds), and service dogs. All three subgroups are represented in this study. Service dogs include mobility assistance dogs, alert service dogs (for the detection of the imminent onset of a medical condition), response service dogs (for the provision of safety during or after a medical episode) and psychiatric service dogs (for the mitigation of a mental health disability, such as autism or PTSD). Emotional support dogs, which are not trained like ADs but provide support by being present, or therapy dogs, which are trained pet dogs, are not included in this system and therefore may not have the same rights as ADs regarding public access [7,8].

In 2016, a UN agreement on the rights of people with disabilities took effect. It states that a person with disabilities should be treated as any other person, that they should have access to mobility aids, and that facilities for the general population should be available to them, in a manner and at a time of their choice [9]. Subsequently, changes were made to Dutch legislation. It clearly states that effective adjustments should be embedded for people with disabilities, including at least the admittance of ADs [10].

This pilot study focuses on the presence of bacteria on the foot pads of ADs and the shoe soles of their users from the Enterobacteriaceae family as a sentinel of general hygiene—this family of bacteria is usually investigated when evaluating food safety and healthcare facility hygiene [11,12]—and *Clostridium difficile*, an important causative agent of diarrhoea, especially in hospitals. Other relevant bacteria for a hospital environment, such as methicillin-resistant *Staphylococcus aureus* (MRSA), vancomycin-resistant enterococci (VRE), and also extended-spectrum beta-lactamase (ESBL) *Escherichia coli*, are rarely found on dogs. *C. difficile*, on the other hand, are found on dogs on a more regular basis [13]. A study conducted in Slovenia on the presence of *C. difficile* showed that these bacteria are found in higher quantities on footwear than on the foot pads of pet dogs [14].

Previous research has been conducted on the microbiological composition of dog fur and paws. A single sample taken from fur yielded a significantly lower prevalence of the Enterobacteriaceae family than a single sample taken from foot pads; only one in 20 dogs carried members of the Enterobacteriaceae family on its fur. In the same study, material collected from the dog’s sleeping place rendered a flea prevalence of 7%. Specimens of the *Cheyletiella* mite and the roundworm *Toxocara* spp. were not found [15]. Because the typical duration of a doctor’s appointment is fairly short (10 to 20 min), the chances of contamination of the environment due to fur is rendered relatively low. The foot pads of dogs can be seen as the equivalent of a person’s shoe soles, and this is why the foot pads were chosen to be the centre of this research. A group of pet dogs (PDs) and their owners were used as a comparison group.

The articles cited in the two previous paragraphs were the basis for this study. Research on the topic of this study is scarce, and so it was designed as a pilot for further investigation.

The aim of this case–control pilot study is (1) to determine if bacteria of the Enterobacteriaceae family and *Clostridium difficile* can be isolated from the paws of ADs and PDs and, if so, to assess the bacterial counts, (2) to compare those numbers to the samples taken from the shoes of their respective users and owners, (3) to investigate if factors can be linked to the amount or presence of bacteria of the Enterobacteriaceae family and if measures are required to reduce the environmental contamination (in a setting such as a hospital) due to ADs, and (4) to assess and to compare the total number of hospital visitors with the number of visitors accompanied by ADs, with the intention to support the results of this study. An additional goal is to gain an overview of the current experiences of AD users regarding admittance to public places (in particular, hospitals) and to suggest possible improvements. This overview is important, because it emphasises the need for more research on this topic (when the majority of the experiences are negative ones).

## 2. Materials and Methods

### 2.1. Materials

Using the results of previous research [15], the sample size was calculated as 100 individuals (50 dogs and 50 humans), to obtain a power of 80%. These numbers were then divided into 25 ADs and their 25 users, and 25 PDs and their 25 owners.

To acquire ADs and their users for this research, the networks of two Dutch official assistance dog foundations were used: The Royal Dutch Guide Dog Foundation and Bultersmekke Assistancedogs. Their members received a newsletter from their respective foundations containing information about the study and they could then enrol themselves by means of an email addressed to the researchers. Information on the study was provided, including any potential risks, the fact that their privacy would be ensured, and that they could withdraw themselves from the study at any moment if they wanted to. Potential participants were sorted by date of enrolment and geographical location, so multiple participants could be visited on one day. PDs and their owners were acquired via the social network of the researchers; this network consisted of family members or friends who owned a dog. The dogs recruited were of a wide variety, including breed, sex, and age. The area covered had a diameter of between 100 and 150 km, was located in the Netherlands (mainly in the middle of the country), and contained urban as well as rural regions.

The visits took place during February and March 2020. Samples from the front paws of the dogs and their owners’ or users’ shoe soles were collected.

In addition, the participants filled out a questionnaire during the visit (hereinafter referred to as “general questionnaire”). The results of the general questionnaire were used to identify possible factors that could be linked to the amount or the presence in general of bacteria of the Enterobacteriaceae family on paws and shoe soles and thus reveal any needed and/or possible measurements to improve hygiene. The factors that were taken into account were dog type (PD or AD), geological location (urban or rural), hypoallergenicity, coat type (long, short, or curly), age in years, sex, neutering (yes or no), vaccination status (per type of vaccination), flea and worm control status, bathing frequency (number of times per year), diet, sleeping place, prohibited areas at home (kitchen, bathroom, or bedroom), weather (wet or dry), and finally the number of walks per day, including time spent outside and locations visited during walks. These locations comprised their own neighbourhood, beaches, forests, designated dog-walking areas, public parks, wastelands, and pastures.

AD users were also asked to answer a few questions about their experiences of admittance to any public place (in particular, hospitals) when accompanied by their AD (hereinafter referred to as “experience questionnaire”). The goals of this questionnaire were to show the urgency of this study and to determine if the situation had changed since previous research conducted by The Royal Dutch Guide Dog Foundation on the admittance of ADs, as described in the Introduction.

Ten Dutch (university) hospitals were approached and asked about their opinion, protocols, and regulations regarding ADs (hereinafter referred to as “hospital questionnaire”). Finally, visitor numbers of hospitals and the percentage of AD users among those numbers—to put them into perspective—were investigated using available data online and personal communication with hospitals, AD users, and AD training organisations.

### 2.2. Methods

#### 2.2.1. Coding

A code, consisting of three parts, was given to each human–dog couple: the first part indicated whether the pair was an AD–user couple (AD) or a PD–owner couple (PD), the middle part was a number from 1 to 25, and the last part identified either the dog (D; dog) or the human (O; owner) within the couple. For example, the first AD–user couple would be assigned the code AD-01, with AD-01-D being the AD and AD-01-O being the AD user. The first PD–owner couple would be assigned the code PD-01, with PD-01-D being the PD and PD-01-O being the PD owner.

#### 2.2.2. Sampling and Preservation

The participants were asked to walk their dogs for 15–30 min, prior to sampling, taking a route which they would normally take. Every participant was asked to wear shoes that they would normally wear to shops, restaurants, hospitals, or on public transport, for example. Within 30 min of the walk, both shoes and the pads and toe webbing of both front paws were sampled by a researcher, who wore sterile gloves. The front paws were preferred over the back paws, because the number of bacteria of the Enterobacteriaceae family is believed to be larger on the front paws [16] (possibly due to being used for digging, among other things) plus they are more accessible since dogs are often accustomed to giving a paw. Their surfaces were swabbed using premoistened Polywipe™ (premoistened with 7 mL phosphate buffer; Medical Wire & Equipment, Corsham, UK) sponge swabs; four sponges per human–dog couple were needed. The sponges were stored in stomacher bags (BagPage, Interscience, Saint Nom, France). The sponge used to swab the right shoe or paw was designated to be further processed for the demonstration and quantification of bacteria of the Enterobacteriaceae family, and the sponge used to swab the left shoe or paw was designated to be further processed for the demonstration of *Clostridium difficile*. The samples were placed in a cooler (approximately 5 to 10 °C) and were processed the same day, within one to six hours.

The Polywipe™ sponges were already shown to be effective for the recovery of *C. difficile* from the environment [14,17]. Prior to this research, a proof of principle was conducted for the recovery of bacteria of the Enterobacteriaceae family. This was done by swabbing the sole of a shoe, which was contaminated with a solution with a known concentration of *E. coli* bacteria. The method in the following paragraph was used and was effective for the recovery of bacteria of the Enterobacteriaceae family; the concentration of bacteria found was equivalent to that of the solution used to contaminate the shoe sole.

#### 2.2.3. Processing of Enterobacteriaceae Samples

First, 93 mL of enrichment medium (peptone–saline solution) was added to the stomacher bags containing the sponges for demonstration and quantification of bacteria of the Enterobacteriaceae family. The contents were homogenised (Bagmixer^®^, Interscience) and a dilution series up to 10^−6^ was executed. Then, 1 mL of every dilution step was placed on Enterobacteriaceae (EB) Petrifilms™ (3M, Saint Paul, MN, USA). The Petrifilms were incubated at a temperature of 37 °C for approximately 24 h. The presence of colonies of the Enterobacteriaceae family was then determined, using colony morphology; colonies of the Enterobacteriaceae family are red and have a yellow halo and/or colony-associated gas bubbles. These colonies were counted.

#### 2.2.4. Processing of *Clostridium difficile* Samples

First, 50 mL of Ringer solution was added to the stomacher bags containing the sponges for demonstration of *C. difficile*. The contents were homogenised. Then, 3 mL of fluid from the bags was transferred to tubes containing 5 mL of enrichment medium (brain/heart infusion broth; produced in-house at IRAS-VPH lab). The tubes were incubated anaerobically at a temperature of 37 °C for approximately 48 h.

Fluid from the tubes was inoculated on Brazier’s agar plates (Oxoid/Thermo Fisher Scientific, Basingstoke, UK), using disposable inoculation loops. The plates were incubated anaerobically at a temperature of 37 °C for approximately 48 h. The presence of *C. difficile* was then determined, using colony morphology; *C. difficile* colonies are grey or white, opaque and flat, have a ground glass appearance and rough, fimbriated edges. The colonies are lecithinase negative and, due to the production of p-cresol, emit a phenolic odour [18]. Further determination was performed by placing the plates under a UV light source (365 nm). *C. difficile* colonies fluoresce yellow to green. For confirmation, a latex agglutination test is needed, but that was not possible during this research considering the budget.

#### 2.2.5. Statistics

Data were collected in Excel spreadsheets (Microsoft, Redmond, WA, USA) and processed using the desktop application of RStudio (Rstudio Inc., Boston, MA, USA). Several tests were conducted. The mean number of colony-forming units (CFUs) of bacteria of the Enterobacteriaceae family was compared between the following groups: human versus dog, PD owners versus AD users, and PDs versus ADs. These comparisons were carried out by means of a *t*-test. To directly compare the dog to its owner or user, paired *t*-tests were performed. The applied hypotheses are shown in Table 1.

Another aim of this study was to investigate the existence of any factors that could be linked to the amount or presence or absence of recovered CFUs of the Enterobacteriaceae family from the right front paws of dogs. For this purpose, (logistic) regression models were set up and executed. Among the eighteen possible factors were age, vaccination status, time spent outside, and weather. Information about these factors was acquired from the general questionnaires.

Finally, the results of the experience questionnaires were reviewed. Chi-squared and proportion tests were performed to determine whether there were any differences in the answers to the questions between AD users with a visible disability and users with an invisible disability, and also between AD users who were in a wheelchair and users who were not. Table 2 presents examples of visible and invisible disabilities. A visual impairment is categorised as a visible disability, because it is often indicated by the use of a red and white cane, and a mobility impairment where a walker or normal cane is used is categorised as an invisible disability, as the elderly also use such aids. Experience shows that, as there are many different AD harnesses, ADs are often mistaken for normal PDs when their user is not visibly disabled or uses an aid that is not typically linked to being disabled. The AD users in wheelchairs did not participate in the sampling, as their shoes are unable to collect contamination from the environment, so they were only asked to fill out the experience questionnaire.

## 3. Results

### 3.1. Sampling Results

In Table 3, the results of the microbiological analysis are shown. The full dataset can be found in Appendix A.

#### 3.1.1. Enterobacteriaceae: Comparing of Groups

During sampling, it was observed that the sponges used for shoe soles were generally looking dirtier than those used for dog paws. It seems that shoe soles are able to collect more material from the environment than dog paws.

The Petrifilms showed that 72% of dogs and 42% of humans returned clean results; “clean” means that zero CFUs of the Enterobacteriaceae family were isolated. When looking at the subgroups, 32% of PD owners, 52% of AD users, 80% of PDs, and 64% of ADs were clean.

Three comparisons were made:Dog versus human. When comparing the mean recovered CFUs between humans and dogs, the two groups differed significantly (*p* < 0.05 and *p* < 0.01). As shown in Table 3, the mean of dogs is lower than that of humans, and thus the general hygiene of dog paws can be considered to be better than that of shoe soles.PD owner versus AD user. When comparing the mean recovered CFUs between PD owners and AD users, the two groups did not differ significantly (*p* > 0.05). The general hygiene of the shoe soles of PD owners and AD users can be considered equal.PD versus AD. When comparing the mean recovered CFUs between PDs and ADs, the two groups did not differ significantly (*p* > 0.05). The general hygiene of the paws of PDs and ADs can be considered equal.

#### 3.1.2. Enterobacteriaceae: Comparing of Couples

A paired *t*-test was conducted, because the observations of a dog and its owner or user are in fact not two separate observations, but dependent ones. A person may not frequent every spot a dog would, and a dog will not always follow its owner, but they take the same route on walks and spend time in the same type of environment. The means of the differences in recovered CFUs between a dog and its owner or user were 1.7868 for the PD group and 0.5496 for the AD group. To calculate these means for each group, the number of recovered CFUs of dogs was subtracted from the number of recovered CFUs of humans.

Two comparisons were made:PD and PD owner couples. The *p*-value that was found for this comparison was very small (*p* < 0.05 and *p* < 0.01), which makes it very unlikely for the mean of the differences to be equal to zero. The general hygiene of the paws of PDs is considered to be better than the shoe soles of their owners.AD and AD user couples. Although the mean recovered CFUs of ADs was found to be lower than that of their users, it was calculated that *p* > 0.05. The general hygiene of the paws of ADs and the shoe soles of their users can be considered equal.

#### 3.1.3. *C. difficile*

*C. difficile* was only found in one sample; however, based on morphology, suspected colonies were found in more samples. This comprised seven of the dog samples, or 14%, and nine of the human samples, or 18%.

#### 3.1.4. *Pseudomonas* spp.

On one Brazier’s agar plate, from which the sample originated from a PD owner’s shoe sole, colonies with atypical morphology for *C. difficile* were present and UV fluorescence occurred. Judging by the smell, these could be *Pseudomonas* spp. colonies. It was decided to further investigate and inoculate some of these colonies onto cetrimide and nalidixic acid (CN) plates, which are specific for *Pseudomonas* spp. This resulted in the growth of colonies fitting with *Pseudomonas* spp. colonies.

### 3.2. Enterobacteriaceae and Possible Factors

The general questionnaire produced a great deal of information and various possible factors that could influence the number and/or presence of recovered CFUs of the Enterobacteriaceae family from dog paws.

The mean bathing frequency was slightly over six (6.1) times per year for ADs and slightly under five (4.8) times per year for PDs. Bathing of a dog was defined as using at least water to wet and clean the whole body of the dog (for example, in a shower or bathtub). This difference was not significant (*p* > 0.05).

Figure 1 shows the vaccination status of the participating dogs. Overall, 100% of ADs were vaccinated with the cocktail vaccine and against leptospirosis, as this is mandatory. Moreover, 64% of PDs had received at least one vaccination; this was significantly different from ADs (*p* < 0.05 and *p* < 0.01). Kennel cough had the lowest vaccination coverage. As rabies is not endemic in the Netherlands, this vaccination is not required, except when the dog travels to a foreign country.

The difference between the proportion of PDs receiving flea control and the proportion of ADs receiving flea control (Figure 2) was significant (*p* < 0.05 and *p* < 0.01). There was no difference in proportions concerning worm control (*p* > 0.05).

As neutering is mandatory for ADs, the proportion of neutered dogs in this group was high, being 84%. The reason why this number is not 100% is that there were three young dogs which were yet to be neutered and one dog which was made an exception. This was a Silken Windsprite, a breed known for having a high risk of complications during or after anaesthesia. In the PD group, 56% were neutered. This was significantly different from the AD group (*p* < 0.05).

Figure 3 shows that while PDs never sleep in or on the beds of their owners, almost half of ADs do. This is because many ADs have to be near their users to do their job, such as detecting epileptic seizures, providing psychological support, or waking them up.

The information about whether or not a dog sleeps on a dog bed or blanket was taken into account, because these beds or blankets may act as an accumulation area for dirt. Subsequently, dogs could gather this additional dirt on their paws when they sleep in such places.

The main component of these dogs’ diets was kibble (Figure 4). Canned food was offered to a small proportion, and raw meat to a slightly larger proportion. Other foods could include snacks or supplements.

A regression test was executed. A multivariable model on the amount of recovered CFUs from dog paws could not yield any factors, as too many CFU counts were zero (which resulted in heteroscedasticity). Therefore, a logistic regression test was performed, to see if any factors could be linked to either the presence or absence of CFUs recovered from dog paws. First, univariable models were set up for each of the 18 possible factors. None of the possible factors were significantly associated. For seven variables, *p* < 0.25 was found (see Table 4). One of these variables had too few individuals per subgroup, and so this variable was omitted (“sleeping place”). The six other possible factors were added to a multivariable model. Association was found for three variables: worm control (not being on worm control), diet (not having “other” as a part of the dog’s diet), and locations visited during walks (“neighbourhood” as a location that was not visited during walks) (*p* < 0.05). Table 5 shows the adjusted odds ratios (ORs) and their 95% confidence intervals (95% CIs) for these associations.

However, the estimates of the variables differed substantially while omitting the variables one by one out of the multivariable model (>15%).

### 3.3. Experiences of AD Users in Public Places (Including Hospitals)

It became clear during this research that, despite lawful access, the participants were frequently halted at public places: almost 81% had experienced this once or several times with their current AD (see Figure 5 for this and more results).

The answers shown in Figure 5 were not significantly different between wheelchair users and non-wheelchair users, but they were distinctly different between the visibly disabled and invisibly disabled (*p* < 0.05).

The most named public places where participants were halted once or more were healthcare facilities (thirteen times), such as hospitals (five times), dentists, doctors’ surgeries (both three times), and pharmacies (two times). Restaurants (ten times) and other food stores (also seven times), such as ice cream parlours, a supermarket, a butcher’s shop, a bakery, a snack bar, and foreign food shops, came in second place. Next were other shops (ten times), such as a hardware store and clothing stores. In fourth place came public transport (six times), such as taxis (four times). Other places that were mentioned were hotels, a playground, a museum, and even the houses of family and friends.

Hygiene was the main reason for these events (named thirteen times), followed by allergies (three times) and inconvenience for other visitors (three times). Another argument that a participant heard was that dogs in general were not allowed in; therefore, they could not make an exception. This again shows that there is a serious lack of knowledge about this subject and the law. Other reasons were that the allowance of the AD would require extra cleaning, that the AD was wearing the wrong harness and was hence not recognised as an official AD (which it was), that they had experienced problems with previous dogs, that a sterile environment must be maintained to the best of their ability, and that dogs are not allowed in the vicinity of food.

A question was included in the experience questionnaire to find out whether or not there were hospitals which generally allowed ADs without any trouble. Overall, 77% of participants had experience with hospitals which generally allowed ADs (Figure 6). They named 29 hospitals, of which 20 were unique.

The differences in answers between wheelchair users and non-wheelchair users, and the visibly disabled and invisibly disabled, were not significant (*p* > 0.05).

Having had experiences of being halted at the entrance of public places, it is imaginable that AD users feel that they cannot take their AD with them to certain places where there may be a low degree of acceptance for these dogs. They were asked if this was something that they were confronted with. Overall, 58% of participants stated that they were. The results per subgroup can be found in Figure 7.

The proportion tests executed on these data showed that there were no significant differences in answers between wheelchair users and non-wheelchair users and between the visibly disabled and invisibly disabled (*p* > 0.05).

Participants said that leaving their AD at home because of a low degree of acceptance could make them feel uncomfortable or uneasy, upset, insecure, and even stressed and not accepted. They felt that it is tiresome to constantly wonder if their dogs were or were not allowed at a certain place. Others explained that it depends on the situation and location, whether it was logical or not that an AD was not allowed (for example, saunas or intensive care departments of hospitals). They said it is also a case of taking other visitors into account.

Participants were asked about possible improvements in public space, public knowledge, and other topics regarding ADs. Their answers are displayed in Table 6.

Participants noted that the largest gain can be made in the public knowledge area and communication between AD users and non-AD users.

### 3.4. Hospitals and Visitor Numbers

#### 3.4.1. Protocols

Unfortunately, only three of the ten approached hospitals responded, and two of these followed common protocols (hereinafter referred to as hospitals A and B, the third hospital being hospital C). This research coincided with the SARS-CoV-2 (COVID-19) pandemic, so it is imaginable that these questionnaires were not a priority.

It is not clear how many hospital visits per year involve an AD. Hospital A stated that they receive around five requests or announcements from AD users regarding bringing their ADs with them.

All of the hospitals agreed on the requirements for an AD: vaccinations and flea and worm control have to be up to date. There were only regular and daily cleaning protocols and no specific ones for after an AD visit. Visible contamination, such as dog hair, faeces, and saliva, is removed from the room and from used materials. Hand hygiene was considered very important. Hygiene checks, such as cleanliness on view or swabbing and culturing, were not being regularly executed; therefore, there were no hygiene limit values in place, such as the maximum allowed number of cultivable CFUs from a certain surface. ADs were not allowed in intensive care units, rooms for aseptic treatments, storage rooms for medical and sterile materials, and food preparation areas. The department of nursing was added to this list by hospitals A and B. Where ADs are allowed, the attending doctor has to give their permission on the admittance of the AD.

There were some differences in how strictly the hospital protocols were applied. Hospital C has no objections to ADs visiting their hospital, providing protocols are followed, and the reason or need to bring their AD is not asked. Hospital A and B require the AD users to announce and register their ADs in advance. They permit ADs to accompany patients to brief treatments (<four hours) or visits to their outpatient clinic. However, ADs are not allowed to accompany their users when they are hospitalised. Moreover, during this period, an AD cannot visit their user. Hospital C does allow ADs during hospitalisation, as long as there is permission from the attending doctor, and the room is cleaned daily.

#### 3.4.2. Visitor Numbers

Using the available data online and following personal communication with hospitals, an overview of visitor numbers in hospitals was constructed. In Figure 8, this overview is visualised. In 2017, there were 126 hospital locations in the Netherlands [19], including eight academic hospitals [20]. One academic hospital claims to have 20,000 people on their premises every day [21].

Hospital A stated that around five ADs visit their facilities every year; for a total of 15,668 unique patients in 2019, this is equal to a percentage of 0.03%. Hospital C receives a few ADs every week; however, their total visitor number is unknown. Hospital C is significantly larger than hospital A.

Following personal communication with AD training organisations and using available data online, an estimation of the total number of ADs in the Netherlands was made. With thirteen accredited organisations, there are over 2200 active human–dog teams. Every year, over 400 new teams are delivered.

Personal communication with 22 of the participating AD users showed that the average number of hospital visits per user is 4.98 times a year. The mode was 4 and the median 3.25. This would mean that the Dutch AD users represent a total of 11,041 hospital visits (outpatient clinic visits and daily admissions) every year. Not every AD user frequents a hospital or takes their AD with them; 18.2% do not. Figure 9 shows a summary of the data.

## 4. Discussion

### 4.1. Sampling

#### 4.1.1. Sampling Results

##### Enterobacteriaceae and General Hygiene

Contrary to many people’s beliefs, this research showed that dog paws in general have better general hygiene than shoe soles. The prejudice of dirty dog paws is not a strange one: the bottom of a dog paw has a large surface area because of their nails, toes, pads, and fur. This may create the opportunity for a large build-up of bacteria.

Besides this, people are often used to taking their shoes off when coming back home, while dogs cannot. However, when visiting a hospital, a shop, or travelling by public transport, shoes are usually not removed. This would make the feet of dogs and humans more equal in these kinds of situations. One potential explanation as to why dog paws are cleaner than shoe soles is that dogs groom themselves, including their paws. This frequency is probably higher than the frequency of shoe cleaning. One study found a grooming frequency of nine to twelve times per hour in dogs kept in group housing [24]. This grooming comprises more than just paw grooming, but it gives a general impression. A fair percentage of research participants reported that they clean the paws of their dogs standardly after a walk. Because shoes are often not worn when at home, they do not need to be cleaned as much. Generally, they are only cleaned when they are visibly dirty.

The grooming can be seen as mechanical cleaning of the paws, and it stimulates blood circulation [25]. Another feasible reason that dog paws are cleaner, also connected to grooming, is the possible antibacterial properties of dog saliva. Canine saliva contains immune [26] and non-immune antimicrobial factors, including lysozyme and salivary peroxidase. The levels of both substances were found to be three times as high as in human saliva [27]. A different study shows that the levels of antimicrobial substances in dog saliva vary between breeds. Samples of the saliva inhibited the growth of *Staphylococcus aureus*, *Pseudomonas aeruginosa*, and *Candida albicans* in vitro, even when concentrations were low [28]. Another study discovered that the saliva of male and female dogs acted bactericidally against the bacteria *E. coli* (which is a member of the Enterobacteriaceae family) and *Streptococcus canis* [29]. This could explain why there were fewer and often zero CFUs recovered from dog paws, in comparison to shoe soles.

Apparently, PD owners and AD users, and PDs and Ads, are equal in general hygiene, as significant differences were not found. The type of human or the type of dog does not influence the general hygiene of their shoe soles or paws, respectively.

To place the found numbers of recovered CFUs in perspective, one could compare these numbers to the number of recoverable CFUs of the Enterobacteriaceae family in dog faeces. On average, these numbers are 10^10^ to 10^11^ CFUs per gram of dog faeces [30]. It would be highly undesirable to enter a public place, especially a hospital, with shoes or paws that are contaminated with faeces. Fortunately, this concentration was never found on either dog paws or shoe soles. Nevertheless, one should check for this kind of contamination when arriving at a public place.

Looking at the human–dog couples, it became clear that, while PD paws have better general hygiene than the shoe soles of their owners, the general hygiene of AD paws and the shoe soles of their users is equal. Thus, the conclusion that “dogs in general have better hygiene” is most likely caused by the better hygiene of PDs, and less so by that of ADs. Why is it that ADs and their users are more equal in general hygiene than PDs and their owners? One theory behind this could be that ADs and their users spend more time together, because a user depends on their AD for support or completing tasks. Away from home, an AD accompanies its user everywhere they go—for example, to supermarkets and hospitals. This is especially the case for guide dogs, as they walk almost the same path as their user when assisting them. This is often not the case for PDs and their owners, as many places do not allow PDs. When at home, a PD and its owner will probably spend more time apart than an AD and its user. These events could explain why ADs and their users are more equal in general hygiene than PDs and their owners.

##### *C. difficile* 

*C. difficile* was only found in one sample, which came from the shoe sole of an AD user. A study conducted in rural and urban Slovenia found 43% of shoe sole samples positive for *C. difficile* and 24% of dog paw samples. The method of this research was mostly adopted into the current research; however, in the Slovenian study, the colonies suspicious for *C. difficile* were confirmed using PCR. In the current research, cultures of seven of the dog paw samples and nine of the shoe sole samples contained colonies that were suspected of being *C. difficile*, because they had most or all characteristics based on colony morphology. Only one sample had colonies that fluoresced under UV light; the rest of the suspicious samples did not. Existing literature does not clarify what these colonies could be. Although the colonies were not confirmed with more sophisticated techniques, it should be noted that Brazier’s agar plates are very specific for *C. difficile* because of the nutrients they contain, the conditions under which they have to incubated, and the way they have to be evaluated. Even though it is not possible with the methods in this study to discern between toxigenic and non-toxigenic *C. difficile*, the bacteria species can act as a sentinel. In future, more extensive research, a confirmation, and specification method, such as PCR, could be used.

##### *Pseudomonas* spp.

*Pseudomonas* spp. were unexpectedly found in one sample when looking for *C. difficile*, being a shoe sole sample from a PD owner. Normally, *Pseudomonas* spp. are aerobic, but some species are able to convert nitrates (including *Pseudomonas aeruginosa*) and, as a result, do not need oxygen [31,32]. This explains why these colonies were able to grow under anaerobic circumstances. In particular, *P. aeruginosa* is a major causative agent for nosocomial infections—for example, urinary and respiratory tract infections—often linked to catheterisation and intubation [33]. Polymicrobial biofilms containing *P. aeruginosa* form on catheters and ventilator tubes, and therefore these bacteria are linked to ventilator associated pneumonia (VAP) [34]. Chronic pneumonia caused by these bacteria is especially dangerous to cystic fibrosis patients [35]. Although it seems that these bacteria are rare on dog paws and shoe soles, as no other plates showed these colonies, this conclusion cannot be drawn from this study and should be further investigated.

#### 4.1.2. Sample Size, Sampling Materials, and Methods

The sample size used for this research was sufficiently large for the comparison of dog versus human. However, because of time and budget limitations, as this was a pilot study, this number was then divided into subgroups, instead of doubled to serve both dog types (AD and PD). Perhaps, when wielding a larger sample size in further investigations, the non-significant differences in means of recovered Enterobacteriaceae CFUs would become significant. As this was a pilot study, a goal was to design a research method on the topic. The next paragraphs will evaluate the method used in this study.

The samples of dog paws were found to be negative (zero recovered CFUs of the Enterobacteriaceae family) more often than the samples of shoe soles. This could imply, again, that dog paws are cleaner than shoe soles. During this research, only diluted samples were inoculated. Therefore, the result “zero recovered CFUs” does not mean that there were not any bacteria of the Enterobacteriaceae family on the paw or shoe; it only shows that the number was below the detection limit (7 × 10^2^). In these cases, either zero CFUs or a number below the confidence limit had grown on the Petrifilms; these numbers were too small and, in such an instance, statistical processing is not possible. Because the method of Polywipe™ sponges and stomacher bags was used, the sponges needed to be suspended in fluid to recover the bacteria sampled from the paws or shoes. A convenient amount of fluid was decided on, to facilitate easy calculation of CFUs and because the surface area of the sponges is fairly large (five by ten centimetres, double-sided). This large surface ensures the recovery of as many bacteria as possible; the fact that the sponges are premoistened also contributes to this goal. This sampling method was therefore deemed to be most effective, also for the recovery of *C. difficile*, especially because the method was already shown to be effective for these bacteria [14,17].

One problem of the used methods is that the different surfaces of dog paws and shoe soles complicate their comparison. It was deemed very difficult to measure the surface of every dog paw, because this surface includes the foot pads, toe webbing, and fur. Merely measuring the bottom’s circumference would not be enough, and also not fair, as this would render a higher concentration of recovered CFUs per cm^2^ than shoe soles. The left and right shoe soles of their owners or users are probably very similar, but the same cannot be said for dog paws. Hind paws are likely to be cleaner than front paws in terms of recoverable CFUs from the Enterobacteriaceae family [16], and so the CFU counts of front paws cannot simply be assumed to be the same as the hind paws. More research is needed in this area. Again, dog paws were found to be negative for Enterobacteriaceae more often than shoe soles, so it would seem that dog paws do indeed have better general hygiene than shoe soles.

At the start of this research, it was not yet clear whether the dogs would tolerate the sampling of their paws. The sponges feel wet and cold to the touch and although they do not have a scent, a dog could still become startled by the smell of the gloves, reminding them of a veterinarian’s office. This was tested during the sampling of the first dog (a PD), and because it tolerated it very well, it was decided this was an appropriate sampling method. The dog was carefully approached from the side while being talked to; it was then allowed to sniff the gloves on the back of the researcher’s hands and was asked to sit down. The researcher gave the command to give paw and waited until the dog obeyed. The foot pads and toe webbing were then carefully swabbed. Afterwards, the dog was rewarded with a treat and petting. Only one dog, which was a PD, did not tolerate the sampling of its paws as well as the other dogs, because it was not used to its paws being handled. Virtually every AD was used to this, so the sampling of their paws did not pose any problems. PD owners were asked beforehand if they thought their PD would tolerate this method, to give rise to successful visits only.

### 4.2. Factors: Linkage to Recovered CFUs

#### 4.2.1. Identified Factors

Based on the executed statistics, three factors could be linked to the presence or absence of recovered CFUs from dogs’ paws: worm control, diet, and locations visited during walks. The first two factors seem difficult to reconcile with the presence or absence of recovered CFUs, especially because raw meat as a part of the diet was not identified as a factor. The ORs from the multivariable model will be discussed in this section, as they hold more value than ORs from univariable models. In this study, the odds of having CFUs recovered from their paws are 15.8 times (or 1480%) higher for dogs that do not visit their neighbourhood during walks than they are for dogs that do. An explanation for this could be that dogs that do not visit their neighbourhood during walks spend more time in locations such as beaches, forests, and parks, which may have higher contamination levels on the ground than the sidewalks and streets in typical neighbourhoods. However, the estimates of the variables differed too much (>15%) while omitting them one by one from the multivariable model, which indicates confounding. It is not clear what the confounders are.

To fully identify factors linked to the amount or the presence or absence of recovered CFUs, it is wise to increase the sample size. In addition, there should be an expansion of certain possible factors. For example, participants were asked how often they bathe their dog in a year; surprisingly, this was not correlated to the absence or presence of recovered CFUs of the Enterobacteriaceae family. This question could be expanded by asking how long ago it was since they bathed their dog. Two other questions that arose during the writing of this report were “How often do participants clean their shoes, and when did they last clean them?”, and “Do participants standardly clean their dogs’ paws after a walk?”. It is advised to ask at least these additional questions during future research on this topic.

#### 4.2.2. Geological Location

Assigning a geological location to a human–dog couple, either urban or rural, was difficult, as a lot of areas are not clearly urban or rural. Some participants lived in the outskirts of a large city, which made their area very similar to the more urban parts of theoretical rural areas.

#### 4.2.3. Hypoallergenicity

Popular literature considers a number of dog breeds to be hypoallergenic, such as the labradoodle, poodle, and Airedale terrier. However, there is a lack of evidence proving the existence of such breeds [36]. Therefore, so-called hypoallergenicity would have no influence on the amount or presence of recovered CFUs, which indeed it did not.

#### 4.2.4. Raw Meat as a Part of a Dog’s Diet

Although raw meat being part of a dog’s diet did not arise as a factor for the general hygiene of their paws, it could still pose a problem. A study found a significantly different faecal microbiome between dogs that were fed on Bones and Raw Food (BARF) diets and dogs that were fed on commercial diets. The BARF diet group had higher abundances of, but was not limited to, *E. coli* and *C. perfringens* in their faeces, compared to commercially fed dogs [37]. Another study showed that raw-meat-based diets (RMBDs) contain several zoonotic bacteria, such as *E. coli*, ESBL, *Listeria* spp., and *Salmonella* spp., and concerning parasites. These pathogens may pose a threat when transmitted to people [38]. A recent study confirmed that the levels of *Salmonella* spp. and *E. coli* in eight commercial RMBDs exceeded the EU standards [39]. This could be especially dangerous to some AD users, as they are considered to be part of a high-risk group (for example, people with chronic diseases). PD owners may also be in this high-risk group, and young children and the elderly always are. When considering all of this, it would not be advisable to feed BARF diets or RMBDs to dogs.

### 4.3. Experiences of AD Users

#### 4.3.1. Results Experience Questionnaires

When faced with the question of whether they had ever been halted at a public place because of their current AD, 81% of the AD users answered that they had. This is often due to lack of information or knowledge, as an explanation by the AD user is usually enough to grant them access; a hard denial is (fortunately) rarer. However, it was noted by many AD users that it takes a lot of effort to keep educating others during their daily routines. This emphasises the need for research on the topic of this study. The participants also had significantly different answers: the proportion that had never been halted was larger for the invisibly disabled than for the visibly disabled. This implies that a larger proportion of the invisibly disabled has not been stopped (yet), compared to that proportion of the visibly disabled. Why is it that a larger percentage of the invisibly disabled has never been halted at a public place, in comparison to the visibly disabled? Looking at the data, this could be caused by the young age of the ADs that have never been halted. The first year of an AD’s life is spent learning the basics of obedience. It depends on the organisation whether the dog already lives with its user or with a foster family. Generally, during the second year of its life, the AD is taught the skills it needs for the aid of its user. Again, it varies per organisation whether the AD already lives with its user or still resides at the organisation’s facilities. Therefore, young ADs have not yet visited many public places. Note that ADs in training, wearing their harness, are also allowed to enter public places.

A number of participants said that they often or always call the place they want to visit with their AD in advance, to announce their intended visit, or immediately walk up to an employee at a store, for example, to explain the situation. It is possible that the invisibly disabled have a stronger tendency to do this, because they feel less understood about the reason for which they make use of an AD. This, and the young ages of the ADs that have never been halted, are possible explanations for the larger percentage of the invisibly disabled that has never been halted, in comparison to the visibly disabled.

The experience questionnaire showed that the most common reason for denying access to ADs and their users is hygiene, or, rather, its violation. For example, restaurant owners may be afraid to lose their licence based on hygiene infringement. They hide behind outdated laws and guidelines, but these no longer apply to ADs. As explained before, the law states that ADs cannot be denied access to public places, including restaurants and hotels. In addition, the hygiene code of the Dutch hospitality industry states that ADs should always be welcome in restaurants, among other places [40]. The Nederlandse Voedsel- en Warenautoriteit (NVWA; similar to the Food and Drug Administration) carries out checks in the hospitality industry. Naturally, they abide by the law, and so their statement emphasises that ADs are allowed to visit restaurants (and cafes and supermarkets), as long as they do not enter kitchens or storage rooms. The space in front of the counter at bakeries, butchers’ shops, and ice cream parlours is also approved for ADs [41].

Logically, this information (both legal requirements and guidelines) must already have been distributed among most people in the hospitality industry, and yet ADs are still denied access. Participants pointed out that their AD is an extension or even a part of themselves, so the denial of an AD is actually the indirect denial of their user. This is where AD organisations need to intervene. Fortunately, they are making good progress in this area. AD users can contact their organisation about a situation where they were denied access because of their AD. Their organisation will then discuss the situation with the public place in question, to push free access even further. The public places that need most attention in this process are healthcare facilities and restaurants and other food stores.

The results of the experience questionnaire also show that hospitals do not discriminate between the types of AD users (wheelchair users and non-wheelchair users, and the visibly disabled and invisibly disabled). The experiences that participants had over the allowance of their ADs into hospitals were not significantly different. Moreover, most of the participants (77%) had been to a hospital that generally allowed ADs without any trouble. The Netherlands has a total of 120 hospitals. This research showed that at least 20 of these have admitted ADs without any trouble. A larger assessment would be needed to fully understand the current situation, as not all hospitals and geographical areas of the Netherlands were researched. Participants did note that the admittance of their AD was strongly dependent on the hospital employee that they encountered on entering the hospital that day, which would mean that a lot of hospitals do not have clear protocols on the allowance of ADs.

The tendency of participants to leave their AD at home, because of a possible low degree of acceptance for these dogs at certain public places, was similar between wheelchair users and non-wheelchair users and between the visibly disabled and invisibly disabled. Overall, 58% of participants stated that they left their AD at home once or several times, because of this possible low degree of acceptance. This is concerning, as their ADs have important tasks to fulfil and they cannot do so if they are away from their users. Naturally, a proportion of users do not need their AD for every task or situation, but they should be able to decide themselves whether it is necessary to take their AD with them, instead of deciding this based on a possible low degree of acceptance from society. This problem distresses some AD users. Again, it is emphasised that access of ADs to public places needs to be upheld and better facilitated. This can be achieved with the help of organisations.

#### 4.3.2. Improvements Suggested by Participants

A number of improvements were mentioned by the participants in the experience questionnaire. It became clear that most problems are caused by a lack of knowledge and poor communication. When ADs are denied access to a public place, this can often be overcome by an explanation of the situation by the AD user. Civilians often do not know what the AD types are, or how to recognise an official AD. This is why there should be more education and publicity about ADs: about their identification cards, what the AD types are, the wide array of breeds that are used, what their harnesses look like, why ADs are so important to their users, and what the law says about ADs. To facilitate the recognition of official ADs, it would be wise to design a uniform AD harness that can be worn by any certified AD, regardless of which organisation trained them. This harness should show their occupation (assistance dog) and a warning not to distract them, in Dutch and in English, accompanied by icons for people who cannot read (young children, for example). This is a topic that is already being discussed. Stichting Gebruikers Assistentiehonden (Stichting GA; a foundation for AD users) is currently working on the uniform recognition of ADs [42].

As mentioned before, AD users can feel as though there is a low degree of acceptance for their ADs. They do not always feel welcome in a public environment. To ameliorate this, stickers could be of great help. Stickers that welcome ADs into stores, for example, already exist, but there are only few of them and they are often targeted at only a single AD type. Participants indicated that these stickers are highly appreciated. Therefore, it is advised to design a sticker that welcomes all types of ADs to enter, and to widely distribute them, along with informative flyers.

The distraction of an AD by a civilian can give rise to dangerous situations. An AD is still a dog, but with an important task to fulfil—for instance, guiding its blind user. ADs are often trained to behave in “work mode” while wearing their harnesses and are “free” when they are not wearing their harnesses. A working AD should not be distracted, which is why a civilian should fully ignore them. This means no petting, no talking to them or making eye contact, even when approached by the dog. PDs can also be a distraction, which is why they should be kept at a distance, ideally on a lead. Potential ADs are carefully selected to ensure that they can handle the job, and they receive sufficient attention from their users [43]. In addition, they receive the same amount of rest as a PD would [44]. Therefore, being an AD is no case of animal cruelty and there is no need for civilians to make up for these assumed shortcomings.

### 4.4. Results of Hospital Questionnaires

The hospitals that completed the hospital questionnaire did have protocols, but they were not always solid. The researchers advise hospitals to set up straightforward, unambiguous, and complete protocols, which include the criteria for ADs (vaccination status, flea and worm control, identification), the range of allowance (departments, duration), and cleaning schedules. Naturally, these protocols should be known to all hospital staff and they should be in line with the law and current guidelines.

### 4.5. Trifecta of Law, Study Results, and Visitor Numbers

The percentage of AD users to total patients was 0.03% in hospital A. It is estimated that 1800 of the 2200 AD users take their AD with them on a hospital visit. As opposed to the total number of patients in the Netherlands, which is 8.28 million, this accounts for a very small percentage, of 0.02%. Over 11,000 hospital visits (outpatient clinic visits and daily admissions) are accompanied by an AD, calculated roughly. In 2017, there were 1,434,000 daily admissions and 8,358,000 first outpatient clinic visits, which combined totals 9,792,000. This number is probably an underestimation, as it only contains the initial outpatient clinic visits. Consequently, the percentage of visits accompanied by an AD amongst those is 0.11%. This could mean that AD users have a higher average of hospital visits per year than non-AD users. Nevertheless, the contribution of AD users to the number of hospital visits is very low. This is why the researchers do not deem it necessary for hospitals to install additional hygiene measures for ADs. Measures that can be taken are, for example, that every hospital visitor could start wearing overshoes. A more environmentally friendly solution could be shoe brushes or longer doormats to reduce the amount of bacteria on shoe soles. These systems would have to be cleaned regularly, or else they would lose their function. Research was done on the usage of sticky mats in hospitals and a reduction in total isolated colonies from shoe soles was found [45]. Some types of sticky mats may also be helpful in decreasing the amount of dust and *C. difficile* spores coming off shoe soles [46]. Another option could be the use of UV lights. Research suggests that these can be effective against several bacterial species found on shoe soles. It acquired the largest reduction in *E. coli* and *C. difficile*, among others, using a stand-on device with a UVC radiation session of eight seconds (HealthySole Plus, HealthySole^®^) [47]. Such a device could be placed in the entrance area of a hospital.

These options may also be applicable to ADs, except for UV lights, as this is detrimental to the skin. Shoes are already available for dogs, and disposable ones could be easily designed. An objection to this product could be that a dog will not tolerate wearing them. One AD user said that they would not want to put their AD through this, so it is debatable how many AD users would approve of this option. Instinctively, doormats can easily be employed for ADs; however, the sticky mats should not hurt the AD by pulling its fur. Finally, dog paws could be cleaned with (wet) wipes on entering the hospital. Naturally, these wipes cannot contain harsh chemicals and should not emit a strong smell, as to not startle the AD. The current study showed that ADs allow their paws to be handled very well; therefore, this could be a promising option.

In combination with the limited contamination that ADs introduce to their environment, this study further emphasises the justification of the law; legally, ADs are allowed to accompany their users into public places, and logically, this is the right decision.

## 5. Conclusions

This pilot study showed that the general hygiene of dog paws is better than that of shoe soles. This result was mostly caused by the better general hygiene of PD paws in comparison to their owners’ shoe soles, as ADs and their users had comparable levels of general hygiene. An explanation for this conclusion may be that ADs and their users spend more time together in the same environments. *C. difficile* was possibly only found on one AD user’s shoe soles. Future research on dog paw hygiene should utilise a larger sample size to investigate possible factors that could be linked to the number or presence or absence of recovered CFUs of the Enterobacteriaceae family.

The experience questionnaire revealed that 81% of AD users had been denied access with their current AD once or several times, despite the law. This underlines the need for research on the topic of this study. Hygiene was one of the main reasons given. The lack of knowledge of the general public on ADs and the law should be addressed, with the help of AD organisations. It would be wise to design a uniform harness for all AD types, regardless of the related organisation. Hospitals should be encouraged to set up straightforward and unambiguous protocols on the admittance of ADs.

AD users only make up a very small part of the total number of patients and hospital visits in the Netherlands (0.02% and 0.11%, respectively). Combined with the results of this study, additional hygiene measures do not seem necessary and, hence, there is no reason that an AD should be denied access to hospitals.

## Figures and Tables

**Figure 1 ijerph-18-00513-f001:**
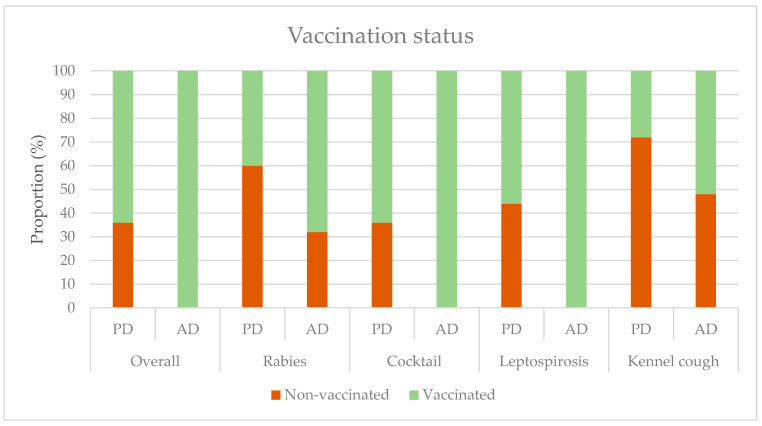
The vaccination status of pet dogs (PD) and assistance dogs (AD): overall (vaccinated = received one or more vaccinations on recommended schedule), rabies vaccination, cocktail vaccination (canine parvovirus, canine distemper virus, and canine adenovirus causing infectious canine hepatitis), leptospirosis vaccination, and kennel cough (canine parainfluenza virus and/or *Bordetella bronchiseptica*).

**Figure 2 ijerph-18-00513-f002:**
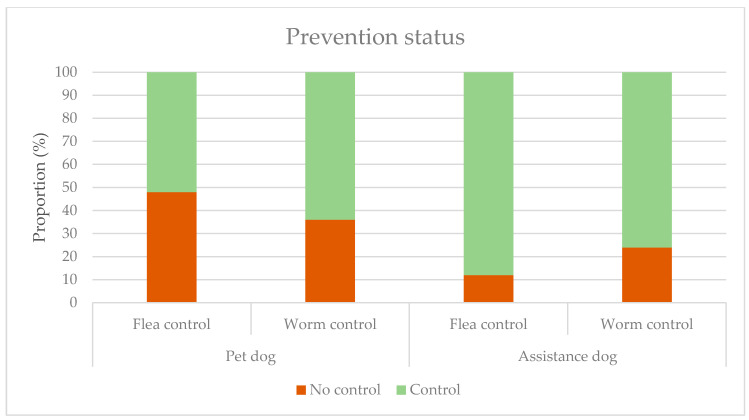
The prevention status of pet dogs (PDs) and assistance dogs (ADs).

**Figure 3 ijerph-18-00513-f003:**
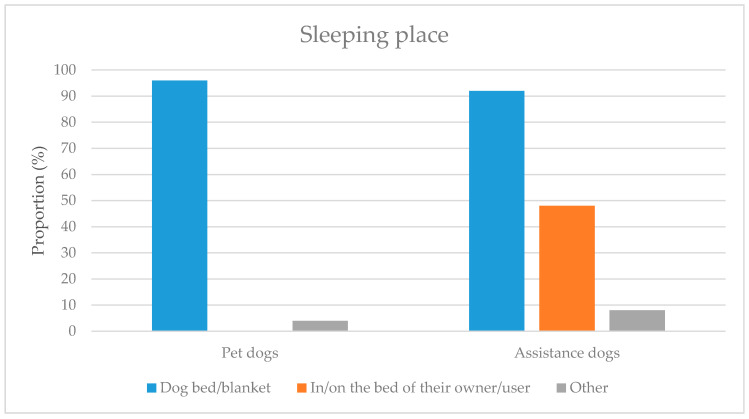
The sleeping places of pet dogs and assistance dogs.

**Figure 4 ijerph-18-00513-f004:**
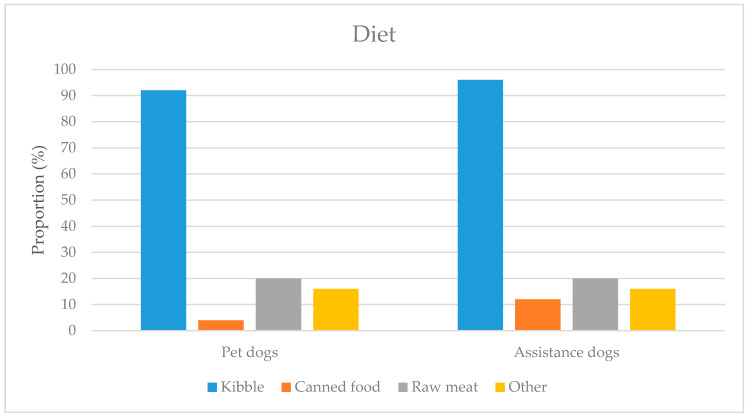
The diets of pet dogs and assistance dogs.

**Figure 5 ijerph-18-00513-f005:**
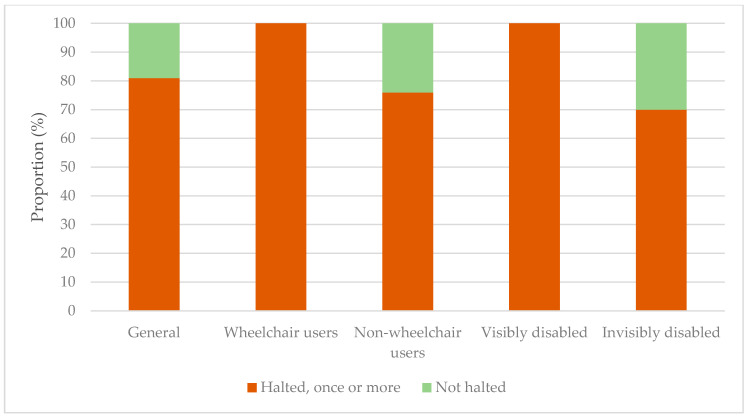
Proportions of participants halted once or more with their current assistance dog at public places: in general and differentiated into subgroups.

**Figure 6 ijerph-18-00513-f006:**
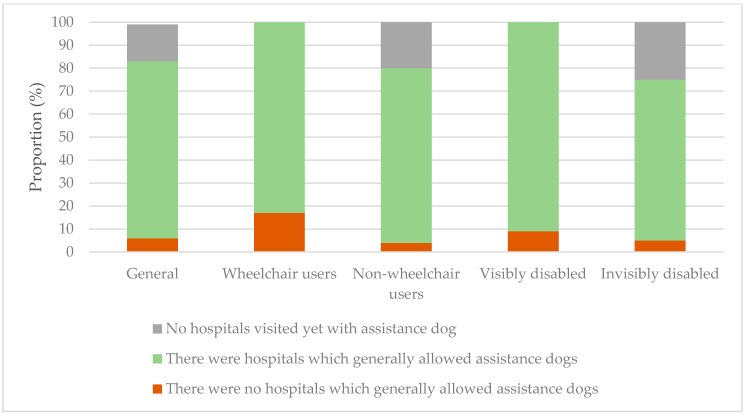
Participants’ experiences of hospital visits with their assistance dogs: in general and differentiated into subgroups.

**Figure 7 ijerph-18-00513-f007:**
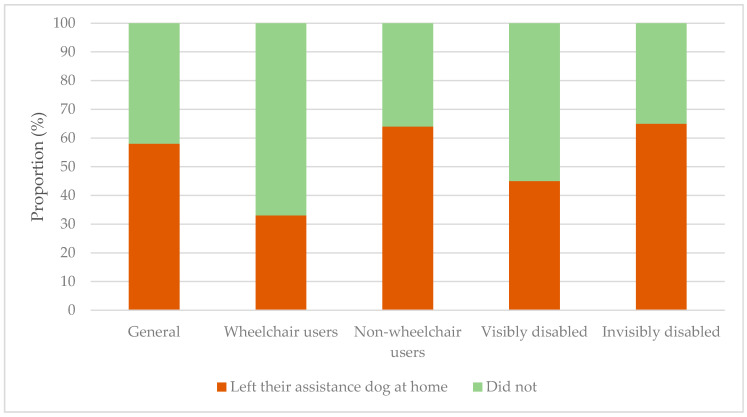
Proportions of participants who left their assistance dog at home because they felt that there was a low degree of acceptance for these dogs at certain places: in general and differentiated into subgroups.

**Figure 8 ijerph-18-00513-f008:**
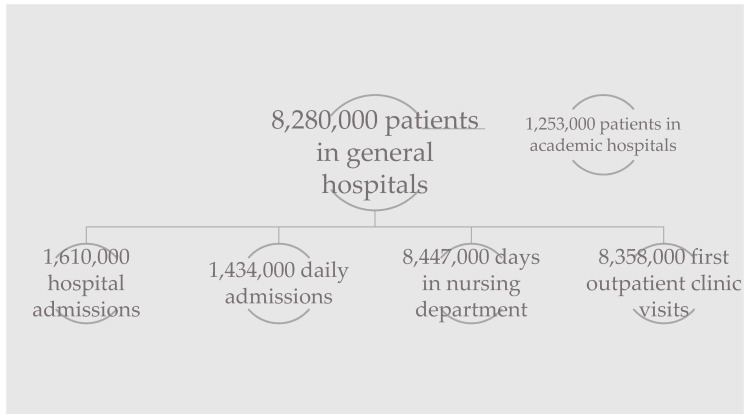
Overview of visitor numbers in Dutch hospitals in 2017 [20,22,23].

**Figure 9 ijerph-18-00513-f009:**
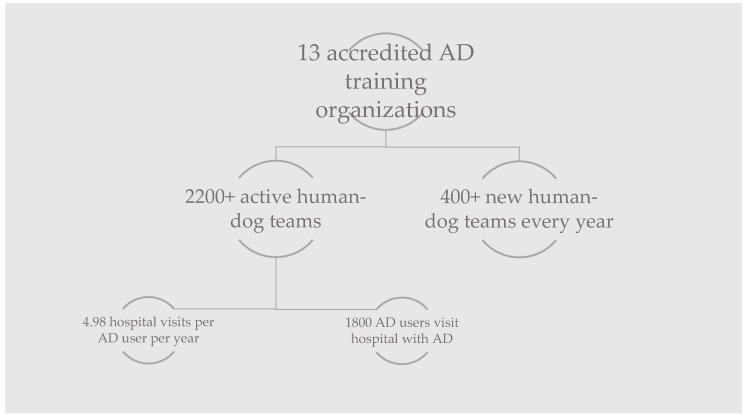
Summary of the numbers regarding assistance dog (AD) training organisations and the AD users of the Netherlands.

**Table 1 ijerph-18-00513-t001:** Hypotheses of the executed *t*-tests and paired *t*-tests, for comparing the mean number of colony-forming units of bacteria of the Enterobacteriaceae family, recovered from dogs’ right front paws and humans’ right shoe soles.

***t*** **-tests**	*Human* vs. *dog*	*H*_0_ = there is no difference between the mean number of colony-forming units of bacteria of the Enterobacteriaceae family recovered from the right front paws of dogs and the right shoe soles of humans.
		*H*_1_ = there is a difference between the mean number of colony-forming units of bacteria of the Enterobacteriaceae family recovered from the right front paws of dogs and the right shoe soles of humans.
	*Pet dog owners* vs. *assistance dog users*	*H*_0_ = there is no difference between the mean number of colony-forming units of bacteria of the Enterobacteriaceae family recovered from the right shoe soles of pet dog owners and the right shoe soles of assistance dog users.
		*H*_1_ = there is a difference between the mean number of colony-forming units of bacteria of the Enterobacteriaceae family recovered from the right shoe soles of pet dog owners and the right shoe soles of assistance dog users.
	*Pet dogs* vs. *assistance dogs*	*H*_0_ = there is no difference between the mean number of colony-forming units of bacteria of the Enterobacteriaceae family recovered from the right front paws of pet dogs and the right front paws of assistance dogs.
		*H*_1_ = there is a difference between the mean number of colony-forming units of bacteria of the Enterobacteriaceae family recovered from the right front paws of pet dogs and the right front paws of assistance dogs.
**Paired *t*-tests**	*Pet dog* vs. *pet dog owner*	*H*_0_ = the mean of the differences, between the number of colony-forming units of bacteria of the Enterobacteriaceae family recovered from pet dogs’ right front paws and their owners’ right shoe sole, is zero.
		*H*_1_ = the mean of the differences, between the number of colony-forming units of bacteria of the Enterobacteriaceae family recovered from pet dogs’ right front paws and their owners’ right shoe sole, is not zero.
	*Assistance dog* vs. *assistance dog user*	*H*_0_ = the mean of the differences, between the number of colony-forming units of bacteria of the Enterobacteriaceae family recovered from assistance dogs’ right front paws and their users’ right shoe sole, is zero.
		*H*_1_ = the mean of the differences, between the number of colony-forming units of bacteria of the Enterobacteriaceae family recovered from assistance dogs’ right front paws and their users’ right shoe sole, is not zero.

**Table 2 ijerph-18-00513-t002:** Examples of visible and invisible disabilities, accompanied by frequently used types of assistance dogs (ADs) [8].

Type of Disability	Examples of Visible or Invisible Disabilities	Examples of Frequently Used Types of ADs
*Visible*	Visual impairment (usage of red and white cane);	Guide dogs;
	Impaired mobility (usage of wheelchair)	Mobility assistance dogs
*Invisible*	Hearing impairment;	Hearing dogs;
	Impaired mobility (usage of walker or normal cane);	Mobility assistance dogs;
	Epilepsy, diabetes;	Alert service dogs, response service dogs;
	PTSD, autism	Psychiatric service dogs

**Table 3 ijerph-18-00513-t003:** Microbiological analysis results: data used for (paired) *t*-tests.

Group	Recovered CFUs * Enterobacteriaceae		*C. difficile* Presence	
	*Mean (Absolute Numbers)*	*Mean (Logarithms)*	*Suspicion*	*UV Light Fluorescence*
*Dogs*	3444	0.9604	7	0
*Humans*	107,893	2.1286	9	1
*Assistance dogs*	1228	1.2000	4	0
*Assistance dog users*	38,364	1.7496	6	1
*Pet dogs*	5660	0.7208	3	0
*Pet dog owners*	177,422	2.5076	3	0

* CFUs: colony-forming units.

**Table 4 ijerph-18-00513-t004:** Unadjusted odds ratios (ORs) and their 95% confidence intervals (95% CIs) for the possible factors in the univariable models with *p*-values < 0.25, linked to the presence or absence of colony-forming units (CFUs) of the Enterobacteriaceae family, recovered from dog paws.

Factors	Number of Dogs		Factors	ORs	95% CIs	*p*-Values
	CFUs Present (*n* = 14)	CFUs Absent (*n* = 36)				
*Dog type: assistance dog*	9 (36%)	16 (64%)	*Being a pet dog*	0.4	0.1–1.6	0.2
*Dog type: pet dog*	5 (20%)	20 (80%)				
*Worm control*	13 (37%)	22 (63%)	*Not being on worm control*	0.1	0.006–0.7	0.05
*No worm control*	1 (7%)	14 (93%)				
*Other elements present in diet*	4 (50%)	4 (50%)	*Not having other elements in their diets*	0.3	0.06–1.5	0.1
*No other elements present (other than kibble, canned food, or raw meat)*	10 (24%)	32 (76%)				
*Neighbourhood as visited location during walks*	11 (24%)	34 (76%)	*Not having “neighbourhood” as a location visited*	4.6	0.7–38.8	0.1
*No neighbourhood as visited location*	3 (60%)	2 (40%)				
*Age (in years): 0–1*	2 (18%)	9 (82%)				
*Age: 2–5*	3 (20%)	12 (80%)	*Age: 2–5*	1.1	0.2–9.9	0.9
*Age: 6–7*	3 (25%)	9 (75%)	*Age: 6–7*	1.5	0.2–13.6	0.7
*Age: 8–13*	6 (50%)	6 (50%)	*Age: 8–13*	4.5	0.7–38.6	0.1
*Vaccinated against rabies*	10 (37%)	17 (63%)	*Not being vaccinated against rabies*	0.4	0.09–1.3	0.1
*Not vaccinated against rabies*	4 (17%)	19 (83%)				
*Sleeping place: bench or dog bed/blanket*	12 (26%)	35 (74%)	*Not having bench or dog bed/blanket as sleeping place*	5.8	0.5–132.4	0.2
*Sleeping place: not bench or dog bed/blanket*	2 (67%)	1 (33%)				

**Table 5 ijerph-18-00513-t005:** Adjusted odds ratios (ORs) and their 95% confidence intervals (95% CIs) for the identified factors in the multivariable model, linked to the presence or absence of colony-forming units (CFUs) of the Enterobacteriaceae family, recovered from dog paws.

Factors	ORs	95% CIs	*p*-Values
*Not being on worm control*	0.04	0.001–0.4	0.007
*Not having other elements in their diets (other than kibble, canned food, or raw meat)*	0.06	0.002–0.5	0.007
*Not having “neighbourhood” as a location visited during walks*	15.8	1.4–339.0	0.04

**Table 6 ijerph-18-00513-t006:** Possible improvements in the infrastructure of public space, public knowledge, and other topics regarding assistance dogs (ADs), as mentioned by AD users.

**Public knowledge**	*For whom?*	Civilians visiting public places.
		Security guards.
		Store personnel.
		People working in the hospitality industry.
		Company owners.
		Healthcare workers.
		Other organisations.
	*About what?*	The fact that every AD or AD in training has an identification card, that shows that it is a certified AD, coming from an official and licensed organisation, and the name of its user.
		Education about hygiene and the impact of ADs on hygiene.
		Education about the reason and need for an AD, and the fact that an AD is something completely different to a pet dog.
		Education about the types of ADs, as people are often only familiar with the guide dog type.
		Dealing with ADs as a non-user: no touching, no talking, no eye contact, ignore the AD (even when it approaches you), keep yourself and your own dog at a distance.
		Education about invisible diseases, as people tend to not recognise these diseases and are often biased.
		Education about the used terms regarding ADs, and the use of standardised terms, set up by Assistance Dogs International (ADI).
		More research on the effect of ADs on their users and corresponding publicity.
		Overall understanding, clarity, and acceptance.
**Infrastructure public space**	*What?*	More space for ADs on public transport; they often have to lie down in the aisle, which can be potentially dangerous, for both people and ADs.
		Not constructing shared places in traffic, especially for people with a visual impairment; they have no orientation and they cannot make eye contact with motorists.
		The availability of an elevator at all times; this is obviously important for wheelchair users, but also when there are only escalators (the fur on dog paws can get caught between the steps).
		Facilitating vacations for ADs and their users; they are often denied in dog-free accommodations. This limits the range of choice and makes the AD user very dependent on certain resorts, hotels, organisations, et cetera. This also applies to transport by taxi.
		It should be noted that revolving doors are an issue for guide dog users.
**Identification and welcoming of ADs**	*What?*	Stickers near the entrances of public buildings, indicating that ADs are welcome. These stickers already exist, but they are only present in small numbers, and frequently targeted at a single type of ADs, most often guide dogs.
		Uniform AD harnesses for every type of AD, regardless of which organisation they are from, which can be recognised by any civilian.
		Education and publicity on these stickers and harnesses, plus information about imitation harnesses of non-official ADs (sometimes even pet dogs).
		Education about communication with AD users: how to handle the situation correctly?

## Data Availability

The data presented in this study are available on request from the corresponding author. The data are not publicly available due to privacy reasons.

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
