# Peer review of "A Pilot Study on the Contamination of Assistance Dogs’ Paws and Their Users’ Shoe Soles in Relation to Admittance to Hospitals and (In)Visible Disability"

_ijerph, 2021, doi:10.3390/ijerph18020513_

Round 1

Reviewer 1 Report

I feel that this is a well-written paper with some valuable information to hospitals and the assistance dog utilizing population that may access the hospitals in their areas. 

The C.difficile isolation continues to be an area of concern and although the authors have added information about the procedure, I wonder if these findings should only be listed as incidental findings. The conclusion in line 798 may want to qualify with "possibly". The findings of this pilot study do need to be expanded and confirmation of C.difficile should probably need to be done.  

The revisions are well done and the paper reads very well. Congratulations.

Author Response

Dear reviewer,

First of all, we would like to thank you again for your interest in our manuscript and your efforts to improve it. We also thank you for your kind words.

Below you’ll find our responses to your last comments.

Line 798 – We added ‘possibly’ to this sentence. Also, we added a line to the C. difficile part in the discussion: we advise to use a confirmation and specification technique for future, more extensive research.

Thank you again for reviewing our manuscript.

Kind regards,

  1. Jasmijn Vos, Joris J. Wijnker and Paul A.M. Overgaauw

Reviewer 2 Report

The paper is ready to the pubblication.

Author Response

Dear reviewer,

We would like to thank you again for your interest in our manuscript and your efforts to improve it. We also thank you for your kind words.

Kind regards,

  1. Jasmijn Vos, Joris J. Wijnker and Paul A.M. Overgaauw

Reviewer 3 Report

Dear Authors,

The manuscript is significatively improved. You performed all suggested modifications or provided reasonable answers to my questions.

In my opinion, the part on Pseudomonas should be omitted because it is out of the study design; but this is my personal point of view, if You think it is important You are free to leave it.

My best Regards

The Reviewer

Author Response

Dear reviewer,

First of all, we would like to thank you again for your interest in our manuscript and your efforts to improve it. We also thank you for your kind words.

Below you’ll find our response to your last comment.

Pseudomonas section – We feel like this could be very important in future research, so that is why we would like to keep this section, also because of the reasons explained in the previous cover letter. Still, we would want to thank you for your suggestion.

Thank you again for reviewing our manuscript.

Kind regards,

  1. Jasmijn Vos, Joris J. Wijnker and Paul A.M. Overgaauw

This manuscript is a resubmission of an earlier submission. The following is a list of the peer review reports and author responses from that submission.

Round 1

Reviewer 1 Report

Revision of manuscript ijerph-974389

Dear Authors,

Your manuscript entitled “Contamination of assistance and pet dogs’ paws and owners’ shoe soles and admittance of public places (hospitals)” faces a very interesting an actual topic that must be discuss and investigate. However, some errors and limitation emerge in microbiological investigation. The work is not well planned, some limit emerged in discussion section and the same Authors admitted some of them.

In particular:

  • In my opinion, the hygienic problems linked to dogs presence in some structures like hospitals can non limited to paws contamination of environment.
  • It is necessary to investigate about the presence and count of more bacteria: Enterobacteriaceae are well hygiene indicator, but other genera must be taken in consideration like Enterococcus or Staphylococcus, and the amount of coli in the totakl Enterobacteriaceae; more in general a total bacterial count must be performed.
  • It has no sense to collect samples after a walking if before paws and shoes were not cleaned and disinfected (see below).
  • A control group must be included: people without dogs (this could be omitted if you performed a disinfection of shoes before walking).

Below a more detailed comment about encountered problems and limitation of the study:

  • Introduction:
    • Authors provided explanations and some supporting references about the choice of looking for Clostridium difficile as the only one pathogen, furthermore, here or in discussion section a deeper discussion about this must be provided; I do not know if this bacterium is the best choice; for examples, in reference [11], focused only on difficile, the percentage of positive dogs is lower than reported in reference [10] (24% vs 80%); Authors based their decision only on 1 manuscript (or they reported only one), but other pathogens must be taken in consideration, or no pathogens must be investigated; furthermore, Clostridium is a spore forming bacteria this must be considered and not all C. difficile isolates are toxigenic (an aspect not investigated by Authors). The same Lefebvre and Colleagues, concluded that “The prevalence of C. difficile in visitation dogs was unexpectedly high.” And the study was conducted in 2004 in Canada. This must be better explained.
  • Material and Methods:
    • Line 106: “time” can not be a limitation in this kind of study, if Authors need more time, the research must go on for more than 2 months. Please, provide a better explanation.
    • Line 145-149: Dear Authors, this part needs a more accurate explanation; if you did not clean and disinfect paws and shoes, haw could you be sure you detect bacteria collected during the walk? If all participant did not start from the same level of contamination this practice (walking) was not necessary; you could simply collect samples directly from animals and humans.
    • Processing of Clostridium difficile samples” section: Line 188-189: this can not be an explanation. Authors did not know if isolated difficile are toxigenic or not and, furthermore, They can not confirm if the isolated colony belong or not to C. difficile, as They own admitted. Results about this part are very ambiguous and not satisfactory. The part about Pseudomonas is not clear and probably out of scope of the investigation; can Authors exclude presence of Pseudomonas for other samples?
    • Many variables were taken in consideration, but no data were reported on the frequencies of dogs were subject to washing (if these information are available and I did not see them, please, apologize me).
  • Results:
    • Table 2: this table did not report information about sampling, but about microbiological investigation.
    • Detailed Results of CFU detected for Enterobacteriaceae should be reported here or as Supplementary material.
    • Lines 231-234: haw could Authors say this if they did not start with clean and disinfected paws and shoes? It is plausible to suppose that dogs, and their paws, are washed and cleaned more frequently than shoes soles (out experiment setting); Authors pic up dirty material from shoes collected by owners before the experiment. How could Authors exclude this?
    • Lines 240-243: see previous comment.
    • Line 241: Table 1?
    • Lines 273-276: these results are difficult to report, discus and consider. Furthermore, these results showed that difficile, or C. difficile alone, is not the best choice as indicator of pathogenic bacteria.
  • Discussion:
    • Line 464-484: I appreciate the discussion provided here by Authors and possible explanation of obtained results; however, I suggest to Authors to be more cautious about the conclusion that paws are cleaner than shoes, for the reasons reported before; Furthermore, although dog saliva has some antimicrobial properties, no recent works or investigation confirm this statement. Finally, Authors planned the sampling of paws and shoes for bacteriological investigation, after a walk of 15-30 minutes; this means they would investigate about the contamination collected during the walk period (with the limitation exposed before); so, considering that paws and shoes sampling were performed immediately after walk, is it possible that dogs had the time for a deeper grooming and paws cleaning?
    • Lines 488-491: please, better explain about this.
    • Lines 506-514: considering the method adopted, Authors must be very careful with conclusion, They can not be sure they isolated difficile and they don’t know if it is toxigenic or not.
    • Lines 515-521: these conclusions are wrong! Authors found accidentally one Pseudomonas while they are looking for Clostridium. They can non exclude other samples are positive for Pseudomonas because they did not search for this bacterium. Pseudomonas are very resistant and obiquitary bacteria, this finding is not unusual. Furthermore, Authors did not characterize the isolated strain, so speculation and conclusion can not be done. “Fortunately, these bacteria seem to be rare on dog paws and shoe soles.”: Authors did not used method for research of Pseudomonas and they can not know this data!
    • Line 532: indeed, to report “zero recovered CFUs” is an error; Authors only included the means and not all the data and this must be solved. 0 CFU can be used for calculation of the mean, but in the result table this datum must be reported as “<700 UFC”!
    • “Sampling methods and materials” section: this section must be reduced or eliminated; if sampling method or bacteriological examination method is a possible limit of the study and no data in literature are available, Authors had to conduct an investigation before to establish the best practices, with appropriate protocols and confrontation that can not be the observations acquired during the study.
    • Lines 566-579: I think it is very difficult to explain the association of CFU count with worm control, but the question is: why did Authors included this in the investigated factors? If there are no plausible reasons for a possible association, there is no sense to ask about this and collect these data, otherwise, possible explanation must be given.
    • Lines 609-725: these data are interesting, but not well integrated with the other part of the study and the title of manuscript; furthermore, this part is very long and dispersive; did You evaluate to convert this part in a standalone manuscript or include it in another kind of study.
  • Discussion:
    • Discussion section must be rewritten taking in consideration previous comments.

I sincerely hope that these suggestions will enhance this manuscript. However, if I have made any errors or misinterpretations, I apologize in advance.

Sincerely

The Reviewer

Reviewer 2 Report

The clarity of the discussion of the findings would be greatly improved if this paper were actually separated into separate papers. If each of the topics remain in this single paper, a more detailed rationale that draws on previous research for each topic should be provided in the introduction of the paper. The sample size of the dog and human pairs, as well as the low response rate from hospitals to report policies, represent major methodological concerns that will limit the potential impact of this paper. The discussion is quite lengthy and really should be condensed to discussing the primary findings in the paper and how they relate to the broader literature on the topic. Improving the specificity of the subheadings in the discussion section would be helpful for guiding the reader through the paper. Additionally, proofreading for grammar and editing for English language is needed throughout the paper.

A few specific comments:

Line 14 – consider using person-first language throughout the text and use “people with disabilities” instead of “disabled people”

Line 19 – I’ve never seen the term “users” associated with the people who benefit from assistance dogs. Maybe “handlers,” “partners,” or “caregivers.”

Line 21 – should use parentheses here – “(e.g., in hospitals)”

Line 27 and Line 34-36 – It is unclear what “visitor numbers” has to do with the scope of this study. Please clarify what this measure consisted of and why it is relevant.

Line 37 – Use of the word “caused” here is not appropriate. Consider using a more appropriate word here such as “demonstrated by”

Line 50 – consider using person-first language throughout the text and use “people who are blind” or “people with a visual impairment”

Line 54 – consider revising to “individuals with invisible disabilities”

Line 62-63 – please revise for person-first language

Line 71-72 - please revise for person-first language

Line 93 – The PDs were a “comparison” group, not a “control” group in this study.

Line 111 – spelling error – “enroll.” Please also provide detail on the consent process for each participant. Was the recruitment of human subjects overseen by an institutional review board? Were participants informed of any potential risks before enrolling?

Line 113 – please provide more detail on what the “network of the researcher” consisted of

Line 114 – “the area covered” – is this to indicate the geographic spread of where each of the participants’ reside? The diameter is provided, but please add detail on what geographic location (city, state, country) these individuals were located in.

Line 129 – what other locations were asked about, besides hospitals?

Reviewer 3 Report

Title: Contamination of assistance and pet dogs’ paws and owners’ shoe soles and admittance of public places (hospitals)

The difficulty of disabled people to bring with them their assistance dog (AD), because of hygiene, is a public problem that has to be solved. This article demonstrates that the general hygiene of AD paws is better than that of people shoe soles, encouraging hospitals to set up straightforward and unambiguous protocols for ADs admittance. The results are original and the methods are adequately described. We recommend publication following a minor revision that I list below:

Comments:

Line 48: substitute “a year” with “per year”;

Line 50: add “to” before “complete”;

Line 52: substitute “by” with “per”;

Line 55: substitute “had” with “have”;

Line 57: substitute “as they wear outside” with “both inside and outside from there”;

Line 59: substitute “utilized” with “used”

Line 71: substitute “took effect” with “has been approved”;

Line 138: substitute “Each human–dog couple was given a code, consisting of three parts” with “A code, consisting of three parts, was given to each human-dog couple”;

Line 149: substitute “for which sterile gloves were worn” with “who wore sterile gloves”;

Line 170: substitute “homogenised” with “homogenized”;

Line 178: substitute “demonstration of C difficile” with “C. difficile detection”

Line 179: “5 mL of enrichment medium” is better to “5 mL enrichment medium”

Line 190: substitute “colonies atypical in morphology” with “colonies with atypical morphology”;

Line 207: "any factor” can be enough;

Line 234: substitute “clean, meaning” with “clean means that”;

Line 235: substitute “when” with “while”;

Line 286: substitute comma to semicolon

In figure 3, a legend is preferred;

Line 323: substitute “CFU” with “CFUs”;

Line 361: substitute “homes” with “houses”;

Line 414: substitute “receive” with “received”;

Line 417: substitute “for after an AD visit, only” with “after an AD visit, but only”;

Line 432: substitute “does allow” with “allows”;

Line 443: substitute “visitor” with “visitors”;

Line 455: substitute “a” with “an”;

Line 444-445: substitute “Hospital A said that about five ADs visit their facilities every year. On their 15,668 unique patients in 2019, that would make a percentage of 0,03%.” with “Hospital A said that about five ADs visit their facilities every year: on a total of 15,668 patients in 2019, that would make a percentage of 0,03%.”;

Line 465: substitute “is not a strange one” with “is not strange”;

Line 468: substitute “to taking” with “to take”;

Line 469: substitute “a” with “an”;

Line 471: add “of” before “why”;

Line 480: remove “of those”;

Line 493: remove When;

Line 496: substitute “Why is it that ADs and their”with “Why ADs and their”;

Line 497: remove “behind this”;

Line 498: substitute “a” with “an” and “their” with “its”;

Line 499: substitute “Outside of their home” with “Outside from home”/ substitute “wherever they go” with “everywhere”;

Line 501: substitute “when assisting them” with “to assist them”;

Line 503: substitute “than” with “from”;

Line 506: substitute “C. difficile was only found in one of the samples, which was a sample from a shoe sole of an AD” with “C. difficile was only found in one sample, which comes from a shoe sole of an AD”;

Line 511-512: substitute “contained colonies that were suspicious of being C. difficile, because they had most or all characteristics based on colony morphology.withcontained suspicious colonies of C. difficile, because of their morphology characteristics”;

Line 513: substitute “light” with “lights”;

Line 524: substitute “The used sample size” with “The sample size used”;

Line 529-530: Substitute “The samples of dog paws came back negative (zero recovered CFUs of the Enterobacteriaceae 529 family) more often than the samples of shoe soles” with “The dog paws samples results more often negative (zero recovered CFUs of the Enterobacteriaceae family)than shoe soles samples.”;

Line 532: remove “any”;

Line 549: substitute “came back negative” with “resulted negative”;

Line 555 and 557: substitute “she” with “it”;

Line 561: substitute “he” with “it” and “his” with “its”;

Line 563-564: “substitute which gave rise to only successful visits” with “to give rise to successful visits only”;

Line 568: substitute “value” with “values”;

Line 570: substitute “than they are for dogs who are” with “than the ones that are”;

Line 573-574: substitute “than they are for dogs who do” with “than the ones who do”;

Line 600: remove “but not limited to”;

Line 607: substitute “and young children and the elderly always are.” with “as well as children and elderly.;

Line 646: remove “yet”;

Line 732: substitute “research” with “investigations”;

Line 733: substitute “to the amount or presence or absence” with “to the amount, presence or absence”;

Line 740: substitute “on the admittance of ADs” with “fort ADs admittance”.

Reviewer 4 Report

Contamination of assistance and pet dogs’ paws and owners’ shoe soles and admittance of public places (hospitals)”

S..Jasmijn Vos, Joris J. Wijnker, Paul A.M. Overgaauw

Manuscript ID: ijerph-974389

1 Nov 20

Thank you for the opportunity to review this interesting and timely paper on equal access for disabled persons. The information in this paper should be of interest to administrative and legal departments of many health care facilities.

Initially, I want to congratulate the authors on a well-written manuscript. There were very few grammatical errors found that should be easily corrected. I did have some areas of concern that I will pose to the authors.

  • In the abstract (line40-41) “Thus, hygiene measures do not seem necessary.” This statement may need to be refined to define specific measures are being addressed.
  • Line 76-77: “… Enterobacteriaceae as a measure of general hygiene” may need to be defined further regarding the foot-pad contamination (fomite?) in contrast to the cited articles use.
  • Line 94: “case-control study” – This appears to be more of an observational study from my basic statistical knowledge. Has a statistician reviewed the experimental design and results?
  • Line106: “power of 80%” – Was this before the groups were divided? If so what is the power of the final groups being evaluated?
  • Lines 177-195, 273-276, 506-521 – The sections discussing difficile are troubling to me because of the lack of confirmation, leaving only subjective assumption of the presence of this important pathogen. I’m not sure if this information should be included in this paper based on the evidence collected.
  • Lines 190-195: Similar to above comment, the evidence for Pseudomonas is subjective and not confirmed. Do you want to include this material?
  • Lines 213-223, 379-380, 654-680: I did not find a itemization of the various disability groups that were discussed in these sections. Did you evaluate statistically the various types of disability regarding access to hospitals and public venues?
  • Line 357,359: Pharmacies is repeated.
  • Line 441,454 and others: “Policlinic” This is not a commonly used name for an “outpatient clinic”. Changing the name may clarify to readers.
  • Line 607-608: Please cite reference if other than opinion.
  • Line 645: “This information…” I assume you mean “legal requirements” or “recommendations” here?
